# Herpes Simplex Virus Mistyping due to HSV-1 × HSV-2 Interspecies Recombination in Viral Gene Encoding Glycoprotein B

**DOI:** 10.3390/v12080860

**Published:** 2020-08-06

**Authors:** Amanda M. Casto, Meei-Li W. Huang, Hong Xie, Keith R. Jerome, Anna Wald, Christine M. Johnston, Alexander L. Greninger

**Affiliations:** 1Division of Allergy and Infectious Diseases, Department of Medicine, University of Washington, Seattle, WA 98195, USA; annawald@uw.edu (A.W.); cjohnsto@uw.edu (C.M.J.); 2Vaccine and Infectious Diseases Division, Fred Hutchinson Cancer Research Center, Seattle, WA 98109, USA; kjerome@fredhutch.org; 3Department of Laboratory Medicine, University of Washington, Seattle, WA 98195, USA; meeili@uw.edu (M.-L.W.H.); hongxie@uw.edu (H.X.); agrening@uw.edu (A.L.G.); 4Department of Epidemiology, University of Washington, Seattle, WA 98195, USA

**Keywords:** HSV-1, HSV-2, HSV typing assays, UL27, glycoprotein B, recombination, genomic variation

## Abstract

Human herpes simplex viruses (HSV) 1 and 2 are extremely common human pathogens with overlapping disease spectra. Infections due to HSV-1 and HSV-2 are distinguished in clinical settings using sequence-based “typing” assays. Here we describe a case of HSV mistyping caused by a previously undescribed HSV-1 × HSV-2 recombination event in UL27, the HSV gene that encodes glycoprotein B. This is the first documented case of HSV mistyping caused by an HSV-1 × HSV-2 recombination event and the first description of an HSV interspecies recombination event in UL27, which is frequently used as a target for diagnostics and experimental therapeutics. We also review the primer and probe target sequences for a commonly used HSV typing assay from nearly 700 HSV-1 and HSV-2 samples and find that about 4% of HSV-1 samples have a single nucleotide change in at least one of these loci, which could impact assay performance. Our findings illustrate how knowledge of naturally occurring genomic variation in HSV-1 and HSV-2 is essential for the design and interpretation of molecular diagnostics for these viruses.

## 1. Introduction

Human herpes simplex viruses (HSV) 1 and 2 cause chronic, incurable infections in 3 billion and 500 million people, worldwide, respectively [1,2]. HSV-1 typically causes orofacial lesions while HSV-2 usually causes genital ulcer disease. However, there is extensive overlap in the spectrum of disease that can be attributed to each virus. The identification of the species of HSV responsible for an infection can provide important diagnostic and prognostic information and is recommended in some clinical contexts by the 2015 Centers for Disease Control and Prevention (CDC) Sexually Transmitted Diseases (STD) Treatment Guidelines [3] and by current Infectious Diseases Society of America (IDSA) Guidelines [4]. HSV speciation or “typing” is performed using commercial test kits or testing assays developed by clinical laboratories. All fourteen FDA-approved HSV typing assays [5] distinguish between HSV-1 and HSV-2 using species-specific probes that recognize genomic DNA or mRNA sequences. As a result, naturally occurring genetic variation in HSV can affect the results of these assays.

Recombination is an important source of genetic variation among HSV samples [6,7,8,9,10,11]. The high rates of recombination observed when distinct HSV strains co-replicate in cell culture, animal models, and human hosts [9,12,13,14] are thought to result from the key role that recombination plays in HSV genomic replication and DNA repair [8,15,16,17,18]. Most recombination in HSV appears to proceed through processes that require sequence homology [17,19,20]. While HSV-1 and HSV-2 are distinct viral species, they likely diverged from a common ancestor only about 6 million years ago and have highly homologous genomes [6,21]. It has long been appreciated that HSV-1 and HSV-2 readily generate recombinant chimeric viruses in vitro [22,23,24,25]. It has also recently been demonstrated that interspecies HSV-1 × HSV-2 recombination contributes to genomic variation among naturally circulating HSV-2 samples [26,27,28]. The extent of the impact of HSV interspecies recombination on HSV genomic variation is still being defined and its relevance for HSV disease in humans remains unknown. Its implications for sequence-based HSV diagnostics also remain unexplored. Here we describe, for the first time, a case of HSV mistyping resulting from a HSV-1 × HSV-2 recombination event in the HSV-2 UL27 gene, which encodes glycoprotein B.

## 2. Materials and Methods

### 2.1. Sample Origin and Ethical Considerations

The HSV sample described in this article, CT_Sample9, was collected in the course of a clinical trial. All trial participants consented to the genetic analysis of their HSV samples. This project was approved by the institutional review board of the University of Washington.

### 2.2. HSV Typing and Genomic Sequencing

HSV samples from all trial participants were sent to the University of Washington Clinical Virology Laboratory for typing using a PCR-based assay that amplifies a portion of the UL27 gene with a different sequence in HSV-1 and HSV-2 [29]. To identify antiviral resistance mutations, species-specific primers were then used to sequence the UL23 (encoding the HSV thymidine kinase) and UL30 (encoding the HSV DNA polymerase) genes (Appendix A).

CT_Sample9 was additionally subjected to whole genome sequencing using a hybridization probe capture technique developed specifically for HSV [30]. A CT_Sample9 consensus sequence for the unique regions of the genome (unique long and unique short, which comprise 78% of the genome) was created via de novo assembly of sequencing reads in Geneious v10 [31].

### 2.3. Sequence Alignment and Recombination Detection

The CT_Sample9 consensus sequence was aligned to HSV-1 Strain 17 (NC_001806.2) and HSV-2 Strain SD90e (KF781518.1) using MAFFT [32]. The portion of this alignment encoding UL27 was scanned for recombination using BootScan as implemented in the recombination detection program (RDP) [33].

### 2.4. Data Availability

The assembled full genome sequence and the raw sequencing reads for CT_Sample9 have been submitted to the National Center for Biotechnology Information (NCBI) under bioproject PRJNA626679.

## 3. Results

### 3.1. Conflicting Assay Results Observed for CT_Sample9

The HSV typing assay used by our clinical laboratory [29] indicated that CT_Sample9 was positive for HSV-1. To assess for antiviral resistance, we next attempted to sequence the UL23 (encoding thymidine kinase) and UL30 (encoding the DNA polymerase) genes of CT_Sample9 using HSV-1-specific primers (Appendix A). However, the genes failed to amplify. A subsequent attempt to amplify these genes using HSV-2-specific primers was successful.

### 3.2. Genome Consensus Sequence for CT_Sample9 Consistent with HSV-2

We next performed whole-genome sequencing on CT_Sample9. De novo assembly of sequencing reads resulted in a 108,998 bp contig and a 16,668 bp contig at an average read depth of more than 2000× that aligned to the unique long and unique short regions, respectively, of both HSV-1 strain 17 and HSV-2 strain SD90e. The CT_Sample9 consensus sequence differed from Strain 17 by 21,829 single nucleotide changes and from SD90e by only 287 changes, demonstrating that CT_Sample9 is HSV-2 and not HSV-1.

### 3.3. HSV-1 × HSV-2 Recombination Event Observed in UL27 in CT_Sample9

The primer and probe sequences for our HSV typing assay are homologous to sequences within the HSV UL27 gene. We reviewed this gene in an alignment of the HSV-1 strain 17, HSV-2 SD90e, and the CT_Sample9 consensus genomes. The assay primers align to loci that are identical in all three genomes (Figure 1). However, the probe region differs between the HSV-1 and HSV-2 references by five single nucleotide changes. The CT_Sample9 consensus genome was identical to the HSV-1 reference sequence in the probe region. Outside the probe region, there were three additional nucleotide positions upstream and five additional nucleotide positions downstream where the HSV-1 and HSV-2 references differed and the CT_Sample9 sequence matched the HSV-1 reference sequence. Two of these changes were non-synonymous (I527V and V529I) (Appendix A). Outside of the region containing these 13 loci, the CT_Sample9 sequence matched the HSV-2 reference at all positions in UL27 where the HSV-1 and HSV-2 references differed. This suggests that the CT_Sample9 genome contains an HSV-1 recombination event of at least 114 bp (maximum length 177 bp) within UL27, which spans these 13 loci. The recombinant region has over 2000-fold read coverage and none of the alleles defining the recombination event were present in less than 98.0% of reads. We also used the program, Bootscan, to examine the UL27 gene in CT_Sample9 for evidence of recombination. The BootScan plot is shown in Figure 2. The p-value for the recombination event, as detected by BootScan, is 1.642 × 10^−17^.

### 3.4. Single Nucleotide Variants Also Found in Primer and Probe Sequences for Typing Assay

As HSV typing assays rely on the homology of primers and probes to viral genomic or mRNA sequences, genetic variation other than HSV-1 × HSV-2 recombination in DNA or RNA sequences targeted by assays also has the potential to interfere with assay results. We scanned 229 publicly available HSV-1 UL27 sequences and found two (0.9%) that had at least one single nucleotide change in our assay’s forward primer target, four (1.8%) that had at least one change in the reverse primer target, and two (0.9%) that had at least one change in the probe target. Seven of the samples with these changes were collected in Seattle. The eighth was collected in Kenya. Four out of 459 publicly available HSV-2 UL27 sequences had target loci that differed from our assay’s HSV-2 probe sequence by the same single nucleotide change. All four of these samples were collected in Seattle. There were no HSV-2 samples with target sequences that differed from either of the primer sequences.

## 4. Discussion

This case illustrates how pathogen genomic variation can affect the accuracy of DNA/RNA sequence-based diagnostics for infectious diseases. Both the recombination event in UL27 and the single nucleotide variants that we identified within the primer and probe target regions are present at low frequencies among sequenced HSV-1 and HSV-2 samples. However, HSV-1 and HSV-2 diversity may not be well-represented by the UL27 and full genome sequences that are currently available, as most of these are from samples collected from a relatively small number of countries [6,12,26,34,35,36]. As an exceedingly small percentage of the thousands of HSV specimens that are typed for clinical purposes every year are sequenced, it is difficult to predict how often mistyping or suboptimal assay performance occurs due to HSV genomic variation. HSV typing assays target many different genomic regions, and the primer and probe sequences for commercial assays are not publicly available, further compounding this uncertainty. Of the 14 FDA-approved commercial HSV typing assays [5], information on the target gene(s) is available for eleven. Seven out of these eleven assays target at least one HSV gene that contains a described HSV-1 × HSV-2 recombination event (Table 1, Appendix A).

This is the first described instance of an HSV-1 × HSV-2 recombination event affecting one of the major HSV glycoprotein-encoding genes. UL27 is a well-studied, essential gene that encodes glycoprotein B, which is critical for viral cell entry. The gene is commonly used as a target for diagnostics and therapeutics, including vaccine candidates. It contains 27 described immune epitopes in HSV-2, one of which is interrupted by the recombination event [37]. As there is no described epitope in the same region in HSV-1, this raises the possibility that the CT_Sample9 recombination event evolved as a means of immunological escape.

In summary, we have described an HSV-1 × HSV-2 recombination event resulting in HSV mistyping. This is the first reported HSV interspecies recombination event in UL27, a gene of diagnostic and therapeutic importance. This case demonstrates that our knowledge of HSV genomic variation remains limited and shows how these limitations can affect the performance of clinical diagnostic tests.

## Figures and Tables

**Figure 1 viruses-12-00860-f001:**
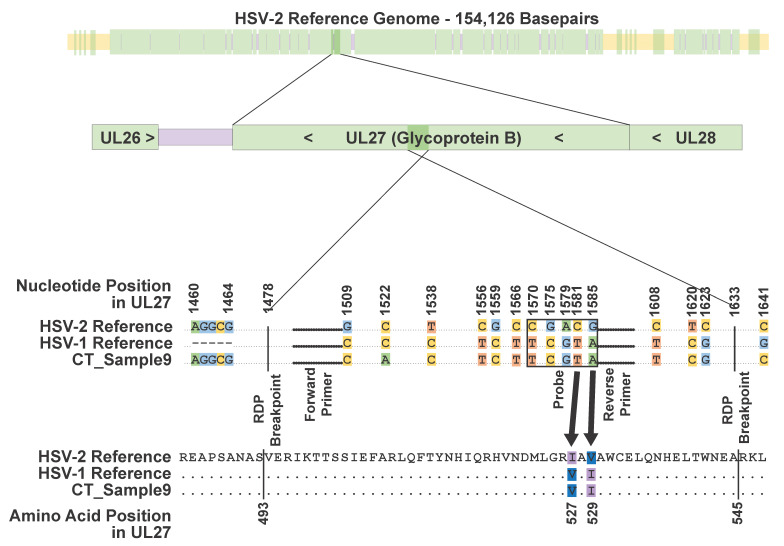
Nucleotides involved in the recombination event in UL27. The top bar shows the location of the UL27 gene (shown in dark colors) within the HSV-2 genome (faded out). Genic regions in the HSV-2 genome are shown in green, while intragenic sequences outside the terminal and internal repeat regions are in purple. Intragenic regions within the terminal and internal repeat regions are in yellow. The second bar shows UL27 in its genic neighborhood with the recombinant region highlighted. UL27 is in reverse orientation in the HSV genome. In the alignment, nucleotides that are identical in all sequences are represented by dots, while nucleotides that differ among sequences are represented by letters. Dots shown in bold represent the locations of the forward and reverse primers, as labeled. The probe region is boxed. Recombination event breakpoints, as identified by the recombination detection program (RDP), are labeled. Non-synonymous differences between HSV-1 and HSV-2 are indicated with arrows pointing to the corresponding amino acid changes.

**Figure 2 viruses-12-00860-f002:**
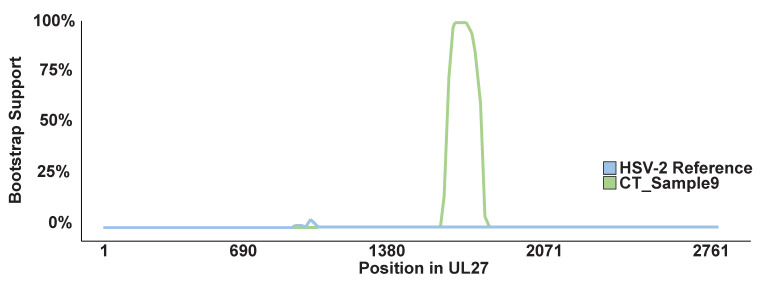
BootScan plot of the HSV-2 reference and CT_Sample9 sequences for UL27. The HSV-1 reference sequence is used as the query sequence.

**Table 1 viruses-12-00860-t001:** Genes targeted by commercial HSV typing assays.

Test	Manufacturer	HSV-1 Target		HSV-2 Target		Site of Known HSV-1 x HSV-2 Recombination Event?
		Gene	Protein Product	Gene	Protein Product	
Aptima	Hologic, Inc.	UL42	DNA Polymerase Processivity Subunit	UL42	DNA Polymerase Processivity Subunit	No
Aries	Luminex Corporation	Not Available		Not Available		NA
IsoAmp	BioHelix Corporation	UL27	Glycoprotein B	UL27	Glycoprotein B	Yes
MultiCode-RTx	EraGen Biosciences, Inc.	UL27	Glycoprotein B	UL27	Glycoprotein B	Yes
BD ProbeTec	BD Diagnostic Systems	Not Available		Not Available		NA
Simplexa	Focus Diagnostics	UL30	DNA Polymerase	UL30	DNA Polymerase	Yes
Amiplivue	Quidel Corporation	Not Available		Not Available		NA
IMDX	Intelligent Medical	US6	Glycoprotein D	UL30	DNA Polymerase	Yes (UL30)
Solana	Quidel Corporation	US7	Glycoprotein I	Intergenic between UL47 and UL48	Intergenic between VP13/14 and VP16	Yes (Intergenic between UL47 and UL48)
Artus	Qiagen	UL30, US6	DNA Polymerase, Glycoprotein D	UL30, US6	DNA Polymerase, Glycoprotein D	Yes (UL30)
SeeGene Anyplex	SeeGene	US6	Glycoprotein D	US6	Glycoprotein D	No
Quidel Molecular Direct	Diagnostic Hybrids	US4	Glycoprotein G	US4	Glycoprotein G	No
Sentosa	Vela Diagnostics	UL30	DNA Polymerase	UL30	DNA Polymerase	Yes
Elite MGB Ingenius	Elitechgroup	US6	Glycoprotein D	US4	Glycoprotein G	No

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
