# Peer review of "Herpes Simplex Virus Mistyping due to HSV-1 × HSV-2 Interspecies Recombination in Viral Gene Encoding Glycoprotein B"

_viruses, 2020, doi:10.3390/v12080860_

Round 1
Reviewer 1 Report
The manuscript “Herpes Simplex Virus Mistyping due to HSV-1 x HSV-2 Interspecies Recombination in Viral Gene Encoding Glycoprotein B” by Casto et al. describes a clinical isolate with interspecies recombination event that resulted in mistyping of HSV. The work clearly presents the HSV-1 x HSV-2 recombination event and the source for the mistyping. These findings are interesting and may influence clinical diagnostics practices.
Minor points
- I think the paper can benefit from further details in the introduction or in the discussion on the HSV recombination process (rate in lab and in clinical isolates).
- Figure 2 is not very informative and can be moved to the supp. information.
- Supp. table 2 is critical for the understanding of the likelihood for mistyping due to interspecies recombination. Thus, I think it should be in the main text.
Author Response
Reviewer #1
The manuscript “Herpes Simplex Virus Mistyping due to HSV-1 x HSV-2 Interspecies Recombination in Viral Gene Encoding Glycoprotein B” by Casto et al. describes a clinical isolate with interspecies recombination event that resulted in mistyping of HSV. The work clearly presents the HSV-1 x HSV-2 recombination event and the source for the mistyping. These findings are interesting and may influence clinical diagnostics practices.
Minor points
I think the paper can benefit from further details in the introduction or in the discussion on the HSV recombination process (rate in lab and in clinical isolates).
We thank the reviewer for this helpful comment. We have added a paragraph to the introduction that provides the reader background on recombination in HSV and on the recent discovery that some naturally circulating HSV-2 viruses carry recombinant HSV-1 in their genomes.
Figure 2 is not very informative and can be moved to the supp. information.
This change was made.
Supp. table 2 is critical for the understanding of the likelihood for mistyping due to interspecies recombination. Thus, I think it should be in the main text.
This change was also made.

Reviewer 2 Report
The observation that HSV 1 and 2 can recombine their genomes is plausible when taking into account earlier wortk from other groups who have shown that HSV DNA replication is biphasic and in the second phase is dependent also on recombination.
However, these fact should be explained to the reader more in detail and relevant literature from the groups of Paul Boehmer, Sandra Weller, Per Elias, Nigel Stow and Bertfired Matz should be cited in the introdcution and included in the discussion.
This comment, however, is minor and does not affect the experimental quality of the study.
Author Response
Reviewer #2 Responses
The observation that HSV 1 and 2 can recombine their genomes is plausible when taking into account earlier work from other groups who have shown that HSV DNA replication is biphasic and in the second phase is dependent also on recombination.
However, these facts should be explained to the reader more in detail and relevant literature from the groups of Paul Boehmer, Sandra Weller, Per Elias, Nigel Stow and Bertfired Matz should be cited in the introduction and included in the discussion.
This comment, however, is minor and does not affect the experimental quality of the study.
We thank the reviewer for this helpful comment. We have added a paragraph to the introduction that states the mechanistic importance of recombination for HSV genomic replication and DNA repair as has been established by the researchers the reviewer notes above among others. We discuss how this leads to frequent recombination among individual HSVs, which acts as a generator of genomic variation among HSV samples.
